# Noise Filtering Method of Digital Holographic Microscopy for Obtaining an Accurate Three-Dimensional Profile of Object Using a Windowed Sideband Array (WiSA)

**DOI:** 10.3390/s22134844

**Published:** 2022-06-27

**Authors:** Hyun-Woo Kim, Myungjin Cho, Min-Chul Lee

**Affiliations:** 1Department of Computer Science and Networks, Kyushu Institute of Technology, 680-4 Kawazu, Fukuoka 820-8502, Japan; kim@ois3d.cse.kyutech.ac.jp; 2School of ICT, Robotics, and Mechanical Engineering, Research Center for Hyper-Connected Convergence Technology, IITC, Hankyong National University, 327 Chungang-ro, Anseong 17579, Korea

**Keywords:** three-dimensional imaging, noise reduction, digital holographic microscopy

## Abstract

In the image processing method of digital holographic microscopy (DHM), we can obtain a phase information of an object by windowing a sideband in Fourier domain and taking inverse Fourier transform. In this method, it is necessary to window a wide sideband to obtain detailed information on the object. However, since the information of the DC spectrum is widely distributed over the entire range from the center of Fourier domain, the window sideband includes not only phase information but also DC information. For this reason, research on acquiring only the phase information of an object without noise in digital holography is a challenging issue for many researchers. Therefore, in this paper, we propose the use of a windowed sideband array (WiSA) as an image processing method to obtain an accurate three-dimensional (3D) profile of an object without noise in DHM. The proposed method does not affect the neighbor pixels of the filtered pixel but removes noise while maintaining the detail of the object. Thus, a more accurate 3D profile can be obtained compared with the conventional filter. In this paper, we create an ideal comparison target i.e., microspheres for comparison, and verify the effect of the filter through additional experiments using red blood cells.

## 1. Introduction

Three-dimensional (3D) imaging technology has been one of the most interesting research topics, recently. Especially, since holography was proposed in 1948 by Dennis Gabor [1], it has been researched in many branches by researchers [2,3,4,5,6,7,8,9,10,11,12,13,14,15,16,17,18,19,20,21,22,23,24]. Holography is a 3D imaging technology that acquires height information of an object by using the coherence of light for phase information of light that cannot be recognized by the human eye or an image sensor. Analog holography technology has disadvantages in that it requires special materials to record images, and it is impossible for modification, compression, and transmission. However, since Joseph Goodman proposed digital holography (DH) in 1967 [25], these disadvantages have been resolved and studied by more researchers. This digital holography technology is being applied to many research fields such as 3D image visualization [3,4], object recognition [5,6], and digital holographic microscopy (DHM) [7,8,9,10,11,12,13,14,15,16,17,18,19,20,21,22,23,24]. Among the mentioned technologies, DHM is widely used in many applications such as microstructure analysis [16,17], microbial research [18,19,20,21], and diagnosis of diseases using cell analysis [22,23] due to capability for obtaining 3D information of micro-objects. In DHM, 3D information of an object can be obtained by using the difference of frequency components between the recorded reference image and the object image. In this processing, windowing one of the sidebands in Fourier domain of the recorded hologram is required to acquire 3D information of the object. The degree of high spatial frequency information of the object shape is determined according to the size of the windowed sideband, that is, as the width of the windowed sideband becomes wider, the detail of the object shape is supplemented. However, the sideband includes the DC spectrum in Fourier domain of the recorded hologram. Therefore, the frequency components of the sideband may not be filtered from the recorded hologram. In addition, when the 3D object information of the hologram is visualized, the information of the DC spectrum included in the windowed sideband becomes noise. In other words, to obtain the detailed shape information of the target which is the high spatial frequency components, windowing range is set to be wide. However, noise may increase in this windowing range because of a lot of the DC spectrum. In contrast, noise may be reduced in the narrow windowing range. However, the high spatial frequency components may be lost in this windowing range due to existence of the only relatively low spatial frequencies. As a result, there is a trade-off between detailed information of an object and noise in DHM. To solve this trade-off, we propose a filtering method which uses a windowed sideband array (WiSA). The proposed method takes an average of 3D information of each windowed sideband which is selected after moving the window at regular intervals from the center of the sideband in Fourier domain of hologram, as including detailed information of an object but reducing noise. Furthermore, since it does not smooth the neighbor pixels of the filtering pixel, it can enhance 3D profile information of the hologram compared with the conventional filtering methods.

This paper has the following structure. First, we mention the concept of the DHM and WiSA in Section 2. Then, we present the experimental setup in Section 3. The experimental results and the discussions of the comparison are shown in Section 4. Finally, we describe the effectiveness of our proposed method and make a conclusion.

## 2. Proposed Method

### 2.1. Principle of the Digital Holographic Microscopy (DHM)

Conventional imaging system can record not the phase information but the intensity of the object. DH is a technology to obtain 3D profile of an object by recording interference pattern between reference and object beams which can record the intensity and phase of the object. Then, the phase of the recorded interference pattern can be obtained, and finally the phase differences between the object image and the reference image can be calculated. The recorded hologram can be written by the following [16]:(1)$$\begin{array}{c} I_{Hol}={\left|R\right|}^{2}+{\left|O\right|}^{2}+R^{*}O+RO^{*} \end{array}$$
where $I_{Hol}$ is the intensity of the recorded hologram, *R* and *O* are the reference and object, respectively. Besides, ${\left|R\right|}^{2}+{\left|O\right|}^{2}$ term is the DC spectrum, $R^{*}O+RO^{*}$ term is the positive or negative sidebands from the Fourier domain, respectively. In addition, $A^{*}$ is the complex conjugate of *A*. Equation (Equation 1) can be rewritten as the following [16]:(2)$$\begin{array}{c} I_{Hol}=I_{R}+I_{O}+R^{*}Oe^{j(\phi_{O}-\phi_{R})}+RO^{*}e^{-j(\phi_{O}-\phi_{R})} \end{array}$$
where $\phi_{O}$ and $\phi_{R}$ are the phase information of the object beam and the reference beam, respectively. Therefore, $\phi_{O}$-$\phi_{R}$ is the phase differences of the recorded hologram. This term can be symbolized as Δϕ, and we can obtain the depth information of the object from Δϕ by using follows [8,10,13,16,24]:(3)$$\begin{array}{c} \Delta \phi \left(x,y\right)=\frac{2\pi }{\lambda }\left(n_{o}-n_{m}\right)h\left(x,y\right), \end{array}$$
(4)$$\begin{array}{c} h\left(x,y\right)=\frac{\Delta \phi (x,y)}{K\times \Delta n}, \end{array}$$
where λ is the wavelength of the illuminated light, and $n_{o}$ and $n_{m}$ are the constant refractive indices of the object and the surrounding medium, respectively. In addition, *h* is the height information of the object, *K* is the wavenumber, and Δn is the refractive index difference between the object and surrounding medium. As a result, by using Equations (3) and (4), we can obtain the height information of the object from the recorded hologram [8,10,13,16,24].

Figure 1a,b show the problem for the image processing of DHM. Figure 1a shows the process of windowing sidebands in Fourier domain. $\pm f_{O}$ in Figure 1a denotes $R^{*}Oe^{j(\phi_{O}-\phi_{R})}$ and $RO^{*}e^{-j(\phi_{O}-\phi_{R})}$ in Equation (Equation 2) for each sideband [13,15,16,24]. That is, it can be seen through Equation (Equation 2) that the phase difference information is contained in the sideband of Fourier domain. Therefore, to obtain the phase difference information of the object, it is necessary to window the area in Fourier domain as depicted in the red square in Figure 1a. Since the detailed shape information depends on the high spatial frequency component of the object in this process, the windowing range has to be wide. However, as this range approaches the origin in Fourier domain as depicted in Figure 1b, DC noise may increase. In addition, it is not easy to avoid this noise because the DC spectrum is distributed throughout the entire Fourier domain unlike that shown in Figure 1a. To solve this trade-off, we propose a DHM filtering method in the next subsection.

### 2.2. Windowed Sideband Array (WiSA) in DHM

Figure 1c illustrates the proposed method. Unlike the conventional DHM, this method windows the sideband array at regular intervals. All windows have the same size and less than the maximum window size. It reduces the phase error of each windowed sideband by calculating the average of the 3D height information obtained from each window. In this case, since all windows have the same size, the detail of the object is maintained. The formula of Figure 1c is written as:(5)$$\begin{array}{c} h_{WiSA}\left(x,y\right)=\frac{1}{\alpha \beta }\sum_{i=1}^{\alpha }\sum_{j=1}^{\beta }\frac{\Delta \phi_{\left(i,j\right)}\left(x,y\right)}{K\times \Delta n},\;\mathrm{for}\;\alpha ,\beta \geq 1,\mathrm{odd} \end{array}$$
where (i,j) is the coordinate of each windowed sideband, α and β are the number of windows in *x* and *y* directions, respectively. The interval between windowed sidebands is set so that the windowed sideband does not exceed the resolution of Fourier domain. To include the windowed sideband of center in Fourier domain, α and β are limited to an odd number. In addition, when α and β are 1, the effect is the same as the conventional method. Thus, they are limited to an odd number greater than 1.

### 2.3. Comparison Method Using Ideal Depth Profile in DHM

In transmissive type DHM, we can obtain the depth information of the object by reconstructing the degree of refraction of the wavefront that occurs in the process of light passing through the object. For this reason, the depth profile by DHM cannot classify the lower part and the upper part of the real object.

Figure 2 describes the difference between real depth information of the object and depth profile by DHM. The object as shown in Figure 2 is a microsphere that is used for calibration of DHM system. In Figure 2, the red solid line, the green dotted line, and the yellow dash-single dotted line illustrate the light path passing through the object, the light path passing through the top of an object or not passing through an object, and the light path passing through the lower part of the object, respectively. As a result, the depth profile by DHM shows accurate height information but does not show the same shape as the object. However, to compare our proposed method with conventional filters, we need an ideal comparison target. Therefore, for comparison, we construct microspheres with a size of 10 µm as an ideal target using the following:(6)$$\begin{array}{c} z_{ICT}=\sqrt{100-4x^{2}-4y^{2}}\times 10^{-6}\left(z_{ICT}\geq 0\right) \end{array}$$
where $z_{ICT}$ is the 3D height data of the ideal comparison target, which is derived to create an ellipsoid centered at the origin with a diameter of 10 on the *x*, *y*-axis and 20 on the *z*-axis by applying the spherical equation. Then, it multiplied by $10^{-6}$ for comparison with the target size of 10 µm, and the final height information data is limited to 0 or greater. As a result, we can obtain a half-ellipsoid with both width and height of 10 µm.

Figure 3 shows an ideal target for depth profile comparison in DHM by Equation (Equation 6). The ideal target as shown in Figure 3 is the identical shape of a rotating body in which the black solid ellipse line as shown in Figure 2b rotates around the center of the vertical axis. For comparison of the results, 2D images in the XY plane are made with maximum and minimum brightness based on the maximum and minimum height values of the original 3D profile without any filters applied. We compare these images with depth profiles of experimental results using the mean square error (MSE), signal-to-noise ratio (SNR), and peak signal-to-noise ratio (PSNR) [26]. We create images with height information of each 3D profile as intensity information and compare the noise of each image by the numerical analysis such as MSE, SNR, and PSNR values [24].

## 3. Experimental Setup

In this paper, we use the modified Mach-Zehnder interferometry to make the fringe pattern as narrow as possible and window the frequency sidebands as wide as possible for including high spatial frequency components of the object.

Figure 4 illustrates the experimental setup for the demonstration of our proposed method, WiSA. The setup uses a 532 nm laser diode module (3 mW output power). In DHM, wavelength of the light source is related to depth resolution, and the lower the wavelength, the more precise depth information can be obtained [16]. However, to avoid the damage of cells such as red blood cells (RBCs) or microorganisms, a green color (532 nm wavelength) laser is used rather than a blue laser (405 nm wavelength). As we mentioned in the previous section, in this experiment, we use microspheres with a size of 10 µm as experimental objects. To avoid overlapping of microspheres, a specimen was made by spreading it thinly on a slide glass. Both the object and the reference beams are magnified by a 40× (0.65 NA) objective lens. In this experiment, since the material of the microspheres we used is polystyrene, 1.5983 which is the constant refractive index of polystyrene for light with a wavelength of 532 nm, is used as the refractive index of the object ($n_{s}$). In addition, the refractive index of the surrounding medium ($n_{m}$) is modified and used based on the height of the object, 10 µm. In this experiment, $n_{m}$ is 1.49. CMOS sensor (Basler, acA2500-14uc, 2590 (H) × 1942 (V) pixel resolution) with a pixel size of 2.2 µm (H) × 2.2 µm (V) is used to capture both reference and object holograms. To obtain the dense fringe pattern by the sampling criteria, the beam splitter in front of the image sensor is tilted.

## 4. Experimental Result

### 4.1. Image Processing of the WiSA

As mentioned earlier, by making the fringe pattern as narrow as possible, the center of $\pm f_{O}$ is located closer to the left and right resolution limits than the DC spectrum. Then, the interval between the window arrays is set as long as possible not to exceed the resolution limit of the recorded hologram. The interference pattern is determined for the center of the sideband of the hologram obtained in this paper to be located between the center of the DC spectrum and the resolution limit. Besides, the maximum limit of the array is calculated as 380 (H) × 380 (V), when α and β are 3, 5, and 7, intervals are 180 pixels, 95 pixels, and 63 pixels, respectively.

Figure 5 shows phase information of reference ($\phi_{R(i,j)}$) and object ($\phi_{O(i,j)}$) holograms obtained by inverse Fourier transform of various windowed sidebands which have the same width centering on the number shown in Figure 1c. In this process, both α and β are 3. As a result, we can obtain the phase difference ($\Delta \phi_{\left(i,j\right)}$) between the phase information of the reference ($\phi_{R(i,j)}$) and the object ($\phi_{O(i,j)}$) obtained at the same location.

Figure 6 shows the phase difference of various windowed sidebands. Then, we can obtain the 3D height profile from this phase difference information using the Goldstein phase unwrapping algorithm [27] and Equation (Equation 5).

### 4.2. Comparison with Conventional Filters

Figure 7 shows the 3D height profile results by various filters such as Gaussian, Median, Average, Wiener, Bilateral and our method. In Figure 7, no filter (NF) method uses the center window of the windowed sideband array. In addition, we apply only α=β case for comparison in this paper. The filter size of each conventional method was modified to obtain the similar SNR/MSE ratio to that of WiSA because the depth information is approximated by the only height information of the object, if the filter size increases. Therefore, we cannot compare the filter effect while maintaining the details. Since it is difficult to compare the 3D profile using only the eye, as mentioned earlier, we compare the effect of various filters by calculating the performance metrics such as MSE, SNR, and PSNR.

Table 1 shows the numerical comparison results by calculating MSE, SNR, and PSNR between the ideal target and 3D profile results by various filters. The numerical results by our proposed method are better than others as the number of α and β increases. This is because the proposed method filters the noise while maintaining the shape details and accurate height information of the object.

### 4.3. Comparison between the Data Processing Time and Filtering Effect

In our proposed method, the data processing time depends on α and β because the data processing amount increases exponentially. Therefore, the optimization of parameters α and β is required for efficiency. To prove the efficiency of our proposed method, we randomly select 15 microspheres and compare them by calculating MSE and PSNR.

Table 2 shows the numerical comparison by calculating MSE and PSNR between the ideal target and the result by WiSA with different amount of data.

For understanding, comparison results as shown in Table 2 are visualized by graph in Figure 8. As shown in Figure 8, both performance metrics of MSE and PSNR show the best efficiency when α and β are 3, considering the processing time. With regardless of the processing time, we can select the largest α and β for the best results. To verify the efficiency of the filter for α and β, we also apply WiSA to the RBCs. The reflective indices of the RBCs ($n_{s}$) and the surrounding medium ($n_{m}$) are used 1.42 and 1.34 (reflective index of the blood plasma), respectively [15].

Figure 9 shows the results by WiSA for the RBCs 3D profile. As shown in Figure 9a, the filtering effect has no significant differences as α and β increase, in contrast, as shown in Figure 9b, we can acquire a more accurate 3D profile of the target as α and β becomes larger even though noise is huge. Therefore, without considering data processing time, it is better to set α and β to a large value.

## 5. Discussion

As a result of comparing the effects of various filters according to α and β and the efficiency of data throughput, considering the processing time, α=β=3 is enough for WiSA because it is more effective than other conventional filtering methods. However, for the best accuracy with regardless of the data processing time, the largest α and β are selected. Our proposed method has shown the strength in terms of getting closer to the shape of an ideal object because it does not use the value of the neighbor pixels of the target pixel to be smoothed. In this paper, we apply the value of α and β only from 3 to 7, but as shown in Figure 8, it is noticed that the larger the value of α and β are used, the closer the effect of the filter becomes the asymptote parallel to the *x*-axis. This means that even if the α and β value increase, there is a limitation to the filter effect. We did not implement the experiment for α=β=9, because the filter effects are similar to each other when the α and β values are 5 and 7. However, we can expect it to show a slightly better filter effect than applying 7.

## 6. Conclusions

In this paper, we have proposed a filtering method to reduce the phase error caused by the DC spectrum in DHM with maintained high spatial frequency information of the target. In addition, we have compared the effect between various filters according to α and β and the efficiency of the data processing amount. The conventional filters compared in this paper are filter algorithms that reduce high-frequency information. As the size of the filter increases, it can have a shape similar to an ideal microsphere with a simple shape. However, our proposed method in this paper has similar effects to filters that reduce high-frequency information while maintaining high-frequency information. Therefore, this is also an advantage of the proposed method. We believe that WiSA can be applied to all other applications based on digital holography.

## Figures and Tables

**Figure 1 sensors-22-04844-f001:**
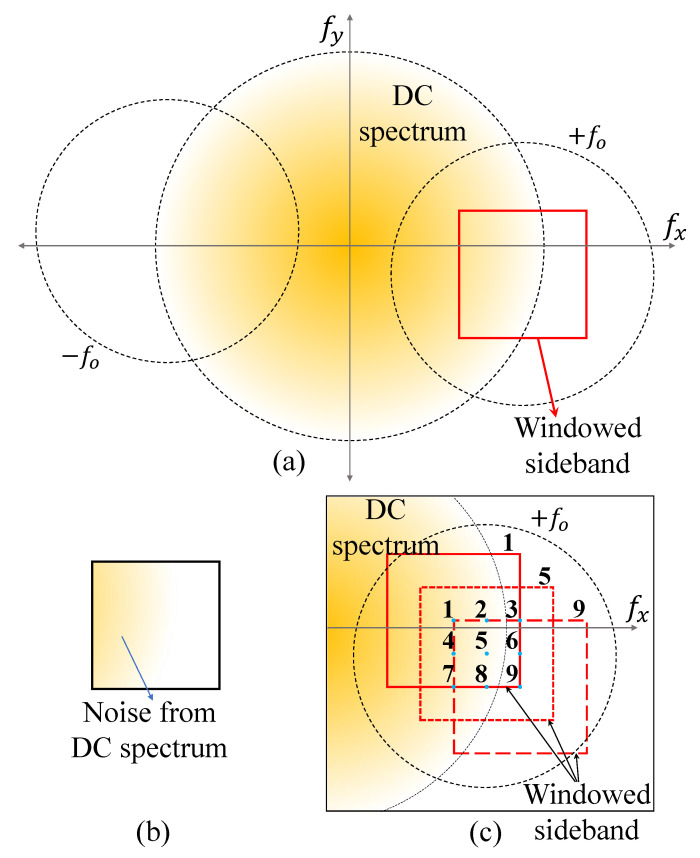
Image processing of DHM (**a**) windowing sidebands from the 2D Fourier domain of the recorded hologram, (**b**) windowed sideband as a red solid square of (**a**,**c**) the proposed method.

**Figure 2 sensors-22-04844-f002:**
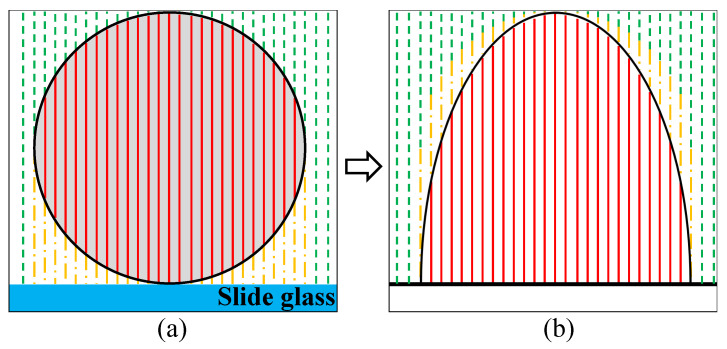
Difference between (**a**) real object (Microsphere) depth information and (**b**) depth profile by DHM.

**Figure 3 sensors-22-04844-f003:**
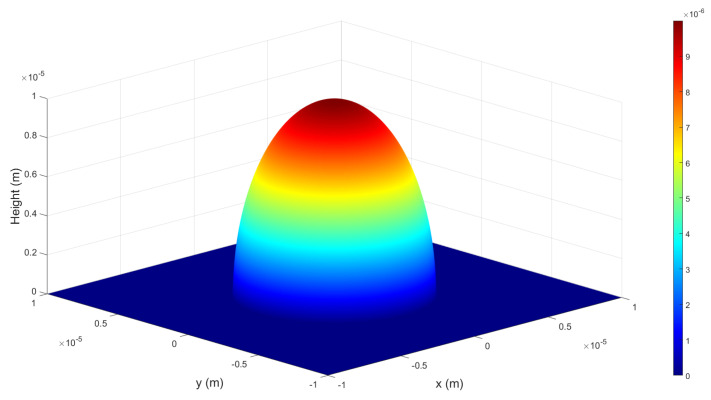
Ideal target for depth profile comparison in DHM.

**Figure 4 sensors-22-04844-f004:**
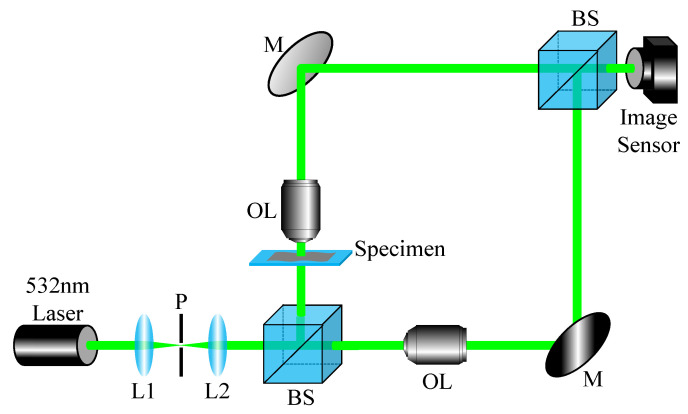
Experimental setup. L: lens, P: pinhole, M: mirror, BS: beam splitter, and OL: objective lens.

**Figure 5 sensors-22-04844-f005:**
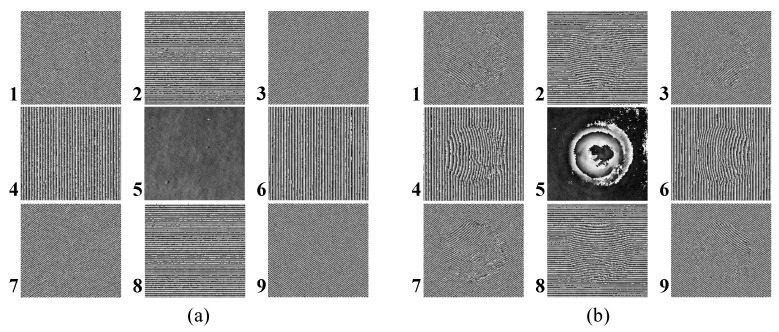
Phase information of (**a**) reference and (**b**) object holograms is obtained by inverse Fourier transform of each windowed sideband.

**Figure 6 sensors-22-04844-f006:**
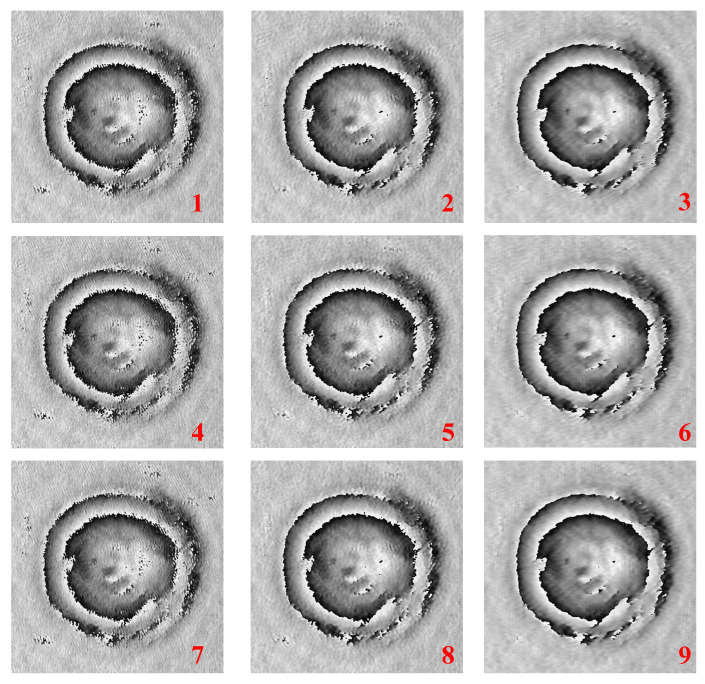
Phase difference of various windowed sidebands.

**Figure 7 sensors-22-04844-f007:**
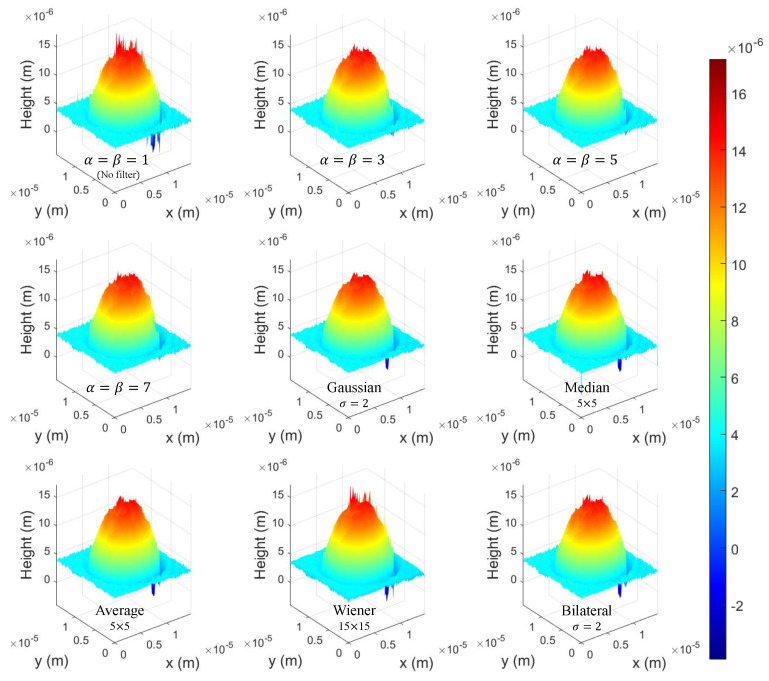
3D height profile results by various filters.

**Figure 8 sensors-22-04844-f008:**
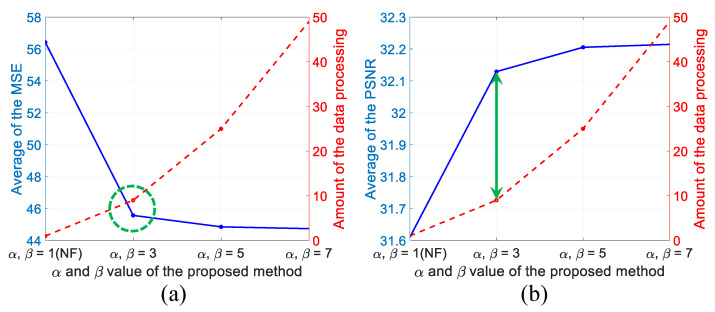
Comparison between the data processing time and filtering effect using (**a**) MSE and (**b**) PSNR. Blue line means result of each metric and red dotted line means amount of the data processing.

**Figure 9 sensors-22-04844-f009:**
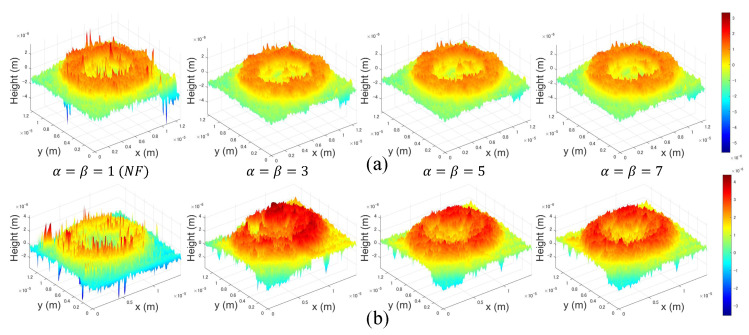
Results by WiSA for the red blood cells (RBCs) 3D profile. (**a**) 3D profile without huge noise and (**b**) 3D profile with huge noise.

**Table 1 sensors-22-04844-t001:** Numerical comparison using MSE, SNR, and PSNR between the ideal target and 3D profile results by various filters.

	NF	α,β = 3	α,β = 5	α,β = 7	Gaussian	Median	Average	Wiener	Bilateral
MSE	14.223	13.687	13.586	13.374	13.809	13.983	13.859	13.880	14.211
SNR	36.601	36.768	36.800	36.868	36.729	36.675	36.714	36.707	36.605
PSNR	6.423	6.590	6.622	6.690	6.552	6.497	6.536	6.529	6.427

**Table 2 sensors-22-04844-t002:** Numerical comparison by calculating MSE and PSNR between the ideal target and the result by WiSA with different amount of data, where SN means sample number.

	MSE				PSNR			
SN	α,β=1(NF)	3	5	7	1(NF)	3	5	7
S1	13.924	13.567	13.298	13.247	36.693	36.806	36.893	36.910
S2	39.762	38.215	37.774	37.250	32.136	32.309	32.359	32.420
S3	39.977	38.580	38.456	38.393	32.113	32.267	32.281	32.288
⋮	⋮	⋮	⋮	⋮	⋮	⋮	⋮	⋮
S14	59.092	58.930	58.903	58.898	30.416	30.427	30.429	30.430
S15	61.618	60.722	60.290	60.414	30.234	30.297	30.328	30.319
avg	56.424	45.572	44.853	44.742	31.610	32.129	32.205	32.215

## Data Availability

Not applicable.

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
