# Peer review of "Noise Filtering Method of Digital Holographic Microscopy for Obtaining an Accurate Three-Dimensional Profile of Object Using a Windowed Sideband Array (WiSA)"

_sensors, 2022, doi:10.3390/s22134844_

Round 1

Reviewer 1 Report

Reviewer’s comments on “Noise filtering method of digital holographic microscopy for obtaining an accurate three-dimensional profile of object using a windowed sideband array (WiSA)” (Sensors-1221316)

Summary:

In this article, the authors propose an interesting filtering scheme using WiSA and make a detailed comparison with the traditional scheme as shown in Table 1. However, the author does not provide a reason for using quantitative standards such as MSE, SNR, and PSNR in manuscript. It is recommended that the author can cite some references those have mentioned quantitative standards used in DHM to highlight the significance of using these quantitative standards for comparison in this study.

The author creates an ideal comparison object and used polystyrene as a sample of microspheres. Comparing the 3D height profile results of Figs. 3 and 7, it is found that the top of each height profile in Fig.7 is not smooth. What is the cause and how can you work out this problem? The axes in Figs 3, 7, and 9 should be marked with unit (m).

Although the originality of this study is not high by the standards of the journal Sensors, the achievement of research should still be of interest to readers in the field of DHM.

The manuscript is well organized and easy to understand. It is useful for the Sensors' readers. I think this manuscript can be considered for publication after response my comments above.

Reviewer 2 Report

Dear author(s), please find some comments on the manuscript ‘Noise filtering method of digital holographic microscopy for obtaining an accurate three-dimensional profile of object using a windowed sideband array (WiSA)’, Manuscript ID: sensors-1771316:

1. What is the three-dimensional (3D) profile? Usually, we have a 3D analysis when considering an n×n array so, respectively, profiles are expressed with 2D data. It should be explained to a regular reader.

2. It was mentioned in the Abstract that the ‘conventional filter’ was compared but, as it is presented, three filter(s) (except the proposed method) were considered, Gaussian, median and average. It should be improved.

3. There is no word, even reference where the ‘Theory’ comes from? Looks like the equations are proposed by the author(s). It would falsely enlarge the novelty presented.

4. From the 3rd section, it is far from a good understanding of what is a proposed novelty that many of the experiment parameters seem to be selected arbitrarily. From that matter, the spectrum of the studies is limited, or the whole proposal as well.

5. The proposed method presents some improvement, however, according to the results from Table 2, differences are (usually) smaller than 1%. Is that useful against general, commonly used (conventional) methods (filters)?

6. From the ‘Conclusion’ section it is far from understanding what the author(s) are trying to convey. Only a few (one, two) sentences present some general, practically useful suggestions. There is more data analysis than a conclusion in this section. It is strongly suggested to present this section with separate, numbered gaps.

From all of the manuscript, (regular) reader seems to be lost, unfortunately. So, respectively, suggested aspects must be significantly improved or must be rejected.

Round 2

Reviewer 2 Report

Dear Authors, according to the review of the revised manuscript ‘Noise filtering method of digital holographic microscopy for obtaining an accurate three-dimensional profile of object using a windowed sideband array (WiSA)’, Manuscript ID: sensors-1771316, it was found appropriately improved and, respectively, can be considered for publication in the quality journal as the Sensors is.

Thank you for taking into consideration the suggested remarks and presenting the full responses, all of them were presented in a required, significant manner.

Concluding, the manuscript can be further processed by the Sensors editorials.